META-RESEARCH ARTICLE

# Relationship between journal impact factor and the thoroughness and helpfulness of peer reviews

Anna Severin[1,2], Michaela Strinzel[3], Matthias Egger [1,3,4]*, Tiago Barros[5], Alexander Sokolov[6], Julia Vilstrup Mouatt[7], Stefan Müller[8]

**1** Institute of Social and Preventive Medicine, University of Bern, Bern, Switzerland, **2** Graduate School for Health Sciences, University of Bern, Bern, Switzerland, **3** Swiss National Science Foundation, Bern, Switzerland, **4** Population Health Sciences, Bristol Medical School, University of Bristol, Bristol, United Kingdom, **5** Faculty Opinions, London, United Kingdom, **6** Clarivate, London, United Kingdom, **7** University of Auckland, Auckland, New Zealand, **8** School of Politics and International Relations, University College Dublin, Dublin, Ireland

* matthias.egger@unibe.ch

**Data Availability Statement:** All relevant data are within the paper and its Supporting Information files. The fine-tuned DistilBERT models, data, and code to verify the reproducibility of all tables and

## Abstract

The *Journal Impact Factor* is often used as a proxy measure for journal quality, but the empirical evidence is scarce. In particular, it is unclear how peer review characteristics for a journal relate to its impact factor. We analysed 10,000 peer review reports submitted to 1,644 biomedical journals with impact factors ranging from 0.21 to 74.7. Two researchers hand-coded sentences using categories of content related to the thoroughness of the review (*Materials and Methods*, *Presentation and Reporting*, *Results and Discussion*, *Importance and Relevance*) and helpfulness (*Suggestion and Solution*, *Examples*, *Praise*, *Criticism*). We fine-tuned and validated transformer machine learning language models to classify sentences. We then examined the association between the number and percentage of sentences addressing different content categories and 10 groups defined by the *Journal Impact Factor*. The median length of reviews increased with higher impact factor, from 185 words (group 1) to 387 words (group 10). The percentage of sentences addressing *Materials and Methods* was greater in the highest *Journal Impact Factor* journals than in the lowest *Journal Impact Factor* group. The results for *Presentation and Reporting* went in the opposite direction, with the highest *Journal Impact Factor* journals giving less emphasis to such content. For helpfulness, reviews for higher impact factor journals devoted relatively less attention to *Suggestion and Solution* than lower impact factor journals. In conclusion, peer review in journals with higher impact factors tends to be more thorough, particularly in addressing study methods while giving relatively less emphasis to presentation or suggesting solutions. Differences were modest and variability high, indicating that the *Journal Impact Factor* is a bad predictor of the quality of peer review of an individual manuscript.

graphs are available at https://doi.org/10.5281/zenodo.8006829. Publons' data sharing policy prohibits us from publishing the raw text of the reviews and the annotated sentences.

**Funding:** This study was supported by Swiss National Science Foundation (SNSF) grant 32FP30-189498 to ME, see https://data.snf.ch/grants/grant/189498) and internal SNSF resources. The funders had no role in study design, data collection and analysis, decision to publish, or preparation of the manuscript.

**Competing interests:** I have read the journal's policy and the authors of this manuscript have the following competing interests: MS and ME were employed by the SNSF and ANS was a PhD student supported by the SNSF at the time of the study. TB, ALS, JVM were employed by Publons (now a part of Web of Science) at the time of the study. SM has declared that no competing interests exist.

**Abbreviations:** CI, confidence interval; COST, European Cooperation in Science and Technology; DORA, San Francisco Declaration on Research Assessment; ESI, Essential Science Indicators.

## Introduction

Peer review is a process of scientific appraisal by which manuscripts submitted for publication in journals are evaluated by experts in the field for originality, rigour, and validity of methods and potential impact [1]. Peer review is an important scientific contribution and is increasingly visible on databases and researcher profiles [2,3]. In medicine, practitioners rely on sound evidence from clinical research to make a diagnosis or prognosis and choose a therapy. Recent developments, such as the retraction of peer-reviewed COVID-19 publications in prominent medical journals [4] or the emergence of predatory journals [5,6], have prompted concerns about the rigour and effectiveness of peer review. Despite these concerns, research into the quality of peer review is scarce. Little is known about the determinants and characteristics of high-quality peer review. The confidential nature of many peer review reports and the lack of databases and tools for assessing their quality have hampered larger-scale research on peer review.

The *impact factor* was originally developed to help libraries make indexing and purchasing decisions for their collections. It is a journal-based metric calculated each year by dividing the number of citations received in that year for papers published in the 2 preceding years by the number of "citable items" published during the 2 preceding years [7]. The reputation of a journal, its impact factor, and the perceived quality of peer review are among the most common criteria authors use to select journals to publish their work [8–10]. Assuming that citation frequency reflects a journal's importance in the field, the impact factor is often used as a proxy for journal quality [11]. It is also used in academic promotion, hiring decisions, and research funding allocation, leading scholars to seek publication in journals with high impact factors [12].

Despite using the *Journal Impact Factor* as a proxy for a journal's quality, empirical research on the impact factor as a measure of journal quality is scarce [11]. In particular, it is unclear how the peer review characteristics for a journal relate to this metric. We combined human coding of peer review reports and quantitative text analysis to examine the association between peer review characteristics and *Journal Impact Factor* in the medical and life sciences, based on a sample of 10,000 peer review reports. Specifically, we examined the impact factor's relationship with the absolute number and the percentages of sentences related to peer review thoroughness and helpfulness.

## Results

### Characteristics of the study sample

The sample included 5,067 reviews from Essential Science Indicators (ESI) [13] research field Clinical Medicine, 943 from Environment and Ecology, 942 from Biology and Biochemistry, 733 from Psychiatry and Psychology, 633 from Pharmacology and Toxicology, 576 from Neuroscience and Behaviour, 566 from Molecular Biology and Genetics, 315 from Immunology, and 225 from Microbiology.

Across the 10 groups of journals defined by *Journal Impact Factor* deciles (1 = lowest, 10 = highest), the median *Journal Impact Factor* ranged from 1.23 to 8.03, the minimum ranged from 0.21 to 6.51 and the maximum from 1.45 to 74.70 (Table 1). The proportion of reviewers from Asia, Africa, South America, and Australia/Oceania declined when moving from *Journal Impact Factor* group 1 to group 10. In contrast, there was a trend in the opposite direction for Europe and North America. Information on the continent of affiliation was missing for 43.5% of reviews (4,355). The median length of peer review reports increased by about 202 words from group 1 (median number of words 185) to group 10 (387). S1 File details the 10 journals from each *Journal Impact Factor* group that provided the highest number of peer review reports, gives the complete list of journals, and shows the distribution of reviews across the 9 ESI disciplines.

**Table 1.  Characteristics of peer review reports by *Journal Impact Factor* group.**

| | Journal Impact Factor group | | | | | | | | | |
|---|---|---|---|---|---|---|---|---|---|---|
| | **1** | **2** | **3** | **4** | **5** | **6** | **7** | **8** | **9** | **10** |
| Median JIF (range) | 1.23 (0.21–1.45) | 1.68 (1.46–1.93) | 2.07 (1.93–2.22) | 2.42 (2.23–2.54) | 2.77 (2.54–3.01) | 3.26 (3.01–3.55) | 3.83 (3.55–4.20) | 4.53 (4.21–5.16) | 5.67 (5.163–6.5) | 8.03 (6.51–74.70) |
| No. of review reports | 1,000 | 1,000 | 1,000 | 1,000 | 1,000 | 1,000 | 1,000 | 1,000 | 1,000 | 1,000 |
| No. of journals | 256 | 224 | 151 | 146 | 183 | 156 | 155 | 129 | 98 | 146 |
| No. of reviewers | 967 | 960 | 969 | 958 | 965 | 973 | 961 | 939 | 970 | 962 |
| No. of sentences (median; IQR) | 9 (4–18) | 11 (6–22) | 12 (5–22) | 13 (6–23) | 14 (7–25) | 14 (7–25) | 16 (8–28) | 17 (8–27) | 16.5 (9–27) | 18 (10–30) |
| No. of words (median; IQR) | 185 (84–359) | 232.5 (116–426) | 225 (104–419) | 256.5 (116–478) | 284.5 (146–506) | 271 (142–495) | 346 (170–581) | 344.5 (176–555) | 350.5 (195–567) | 387 (213–672) |
| Continent of reviewers' affiliation | | | | | | | | | | |
| Asia | 139 | 107 | 163 | 115 | 93 | 135 | 98 | 93 | 80 | 62 |
| Africa | 15 | 14 | 18 | 9 | 5 | 14 | 8 | 6 | 5 | |
| Europe | 119 | 156 | 187 | 190 | 231 | 250 | 268 | 273 | 280 | 241 |
| North America | 97 | 113 | 105 | 153 | 162 | 151 | 191 | 180 | 166 | 213 |
| Central/South America | 61 | 42 | 36 | 25 | 38 | 22 | 22 | 20 | 23 | 10 |
| Australia/Oceania | 50 | 55 | 36 | 46 | 64 | 37 | 26 | 37 | 38 | 52 |
| Missing | 519 | 513 | 455 | 462 | 407 | 391 | 387 | 391 | 408 | 422 |
| Gender of reviewer | | | | | | | | | | |
| Female | 242 | 262 | 261 | 254 | 241 | 211 | 216 | 189 | 260 | 206 |
| Male | 518 | 516 | 478 | 549 | 548 | 551 | 575 | 584 | 543 | 599 |
| Unknown | 240 | 222 | 261 | 197 | 211 | 238 | 209 | 227 | 197 | 195 |

IQR, interquartile range; JIF, Journal Impact Factor.

Continents are ordered by population size.

JIF group defined by deciles (1 = lowest, 10 = highest).

## Performance of coders and classifiers

The training of coders resulted in acceptable to good between-coder agreement, with an average Krippendorff's $\alpha$ across the 8 categories of 0.70. The final analyses included 10,000 review reports, comprising 188,106 sentences, which were submitted by 9,259 reviewers to 1,644 journals. In total, 9,590 unique manuscripts were reviewed.

In the annotated dataset, the most common categories based on human coding were *Materials and Methods* (coded in 823 sentences or 41.2% out of 2,000 sentences), *Suggestion and Solution* (638 sentences; 34.2%), and *Presentation and Reporting* (626 sentences; 31.3%). In contrast, *Praise* (210; 10.5%) and *Importance and Relevance* (175; 8.8%) were the least common. On average, the training set had 444 sentences per category, as 1,160 sentences were allocated to more than 1 category. In out-of-sample predictions based on DistilBERT, a transformer model for text classification [14], precision, recall, and F1 scores (binary averages across both classes [absent/present]) were similar within categories (see S2 File). The classification was most accurate for *Example* and *Materials and Methods* (F1 score 0.71) and least accurate for *Criticism* (0.57) and *Results and Discussion* (0.61). The prevalence predicted from the machine learning model was generally close to the human coding: Point estimates did not differ by more than 3 percentage points. Overall, the machine learning classification closely mirrored human coding. Further details are given in S2 File.

## Descriptive analysis: Thoroughness and helpfulness of peer review reports

The majority of sentences (107,413 sentences, 57.1%) contributed to more than 1 content category; a minority (23,997 sentences, 12.8%) were not assigned to any category. The average number of sentences addressing each of the 8 content categories in the set of 10,000 reviews ranged from 1.6 sentences on *Importance and Relevance* to 9.2 sentences on *Materials and Methods* (upper panel of Fig 1). The percentage of sentences addressing each category are shown in the lower panel of Fig 1. The content categories *Materials and Methods* (46.7% of sentences), *Suggestion and Solution* (34.5%), and *Presentation and Reporting* (30.0%) were most extensively covered. The category *Results and Discussion* was present in 16.3% of the sentences, and 13.1% were assigned to the category *Examples*. In contrast, only 8.4% of sentences addressed the *Importance and Relevance* of the study. *Criticism* (16.5%) was slightly more common than *Praise* (14.9%). Most distributions were wide and skewed to the right, with a peak at 0 sentence or 0% corresponding to reviews that did not address the content category (Fig 1).

Fig 2 shows the estimated number of sentences addressing the 8 content categories across the 10 *Journal Impact Factor* groups. For all categories, the number of sentences increased from *Journal Impact Factor* groups 1 to 10. However, increases were modest on average, amounting to 2 or fewer additional sentences. The exception was *Materials and Methods*, where the difference between *Journal Impact Factor* groups 1 and 10 was 6.5 sentences on average.

Fig 3 shows the estimated percentage of sentences across content categories and *Journal Impact Factor* groups. Among thoroughness categories, the percentage of sentences addressing

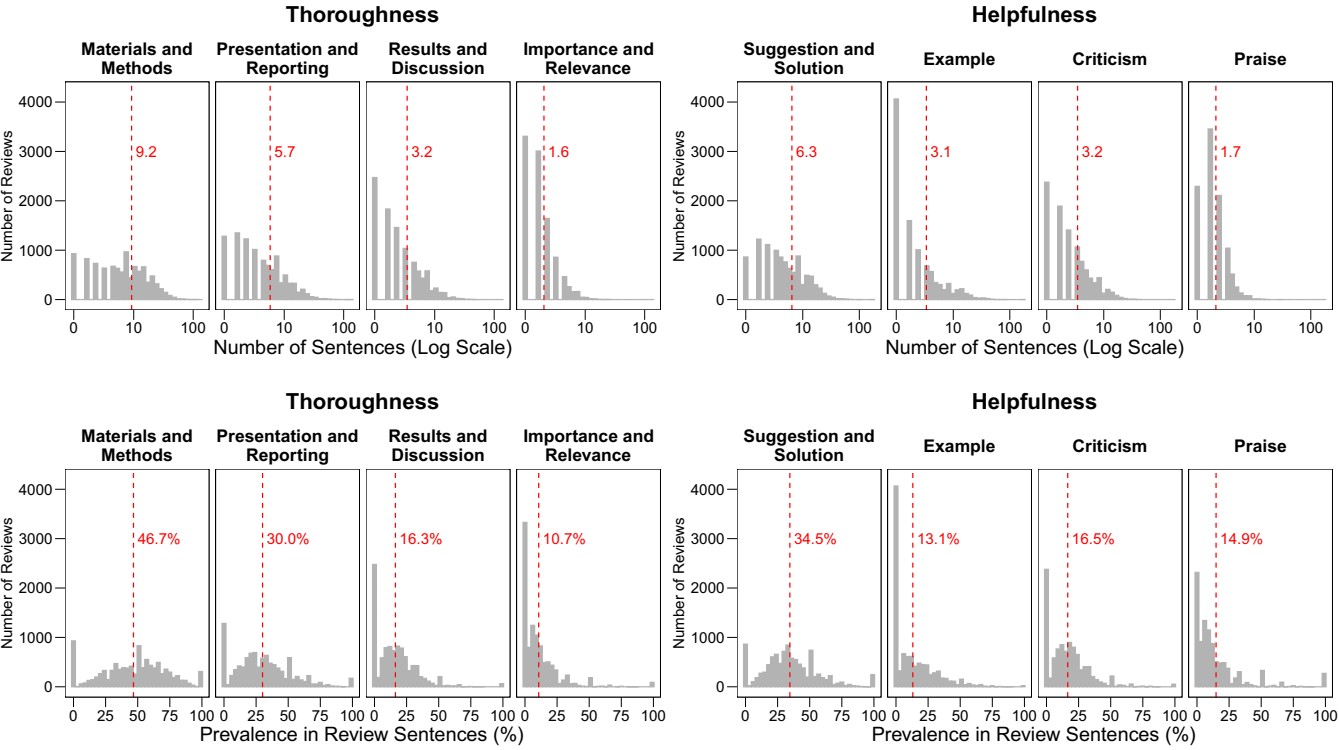

**Fig 1. Distribution of sentences in peer review reports allocated to 8 content categories.** The number (upper panel) and percentage of sentences (lower panel) in a review allocated to the 8 peer review content categories is shown. A sentence could be allocated to no, one, or several categories. Vertical dashed lines show the average number (upper panel) and average percentage of sentences (lower panel) after aggregating them to the level of reviews. Analysis based on 10,000 review reports. The data underlying this figure can be found in S1 Data.

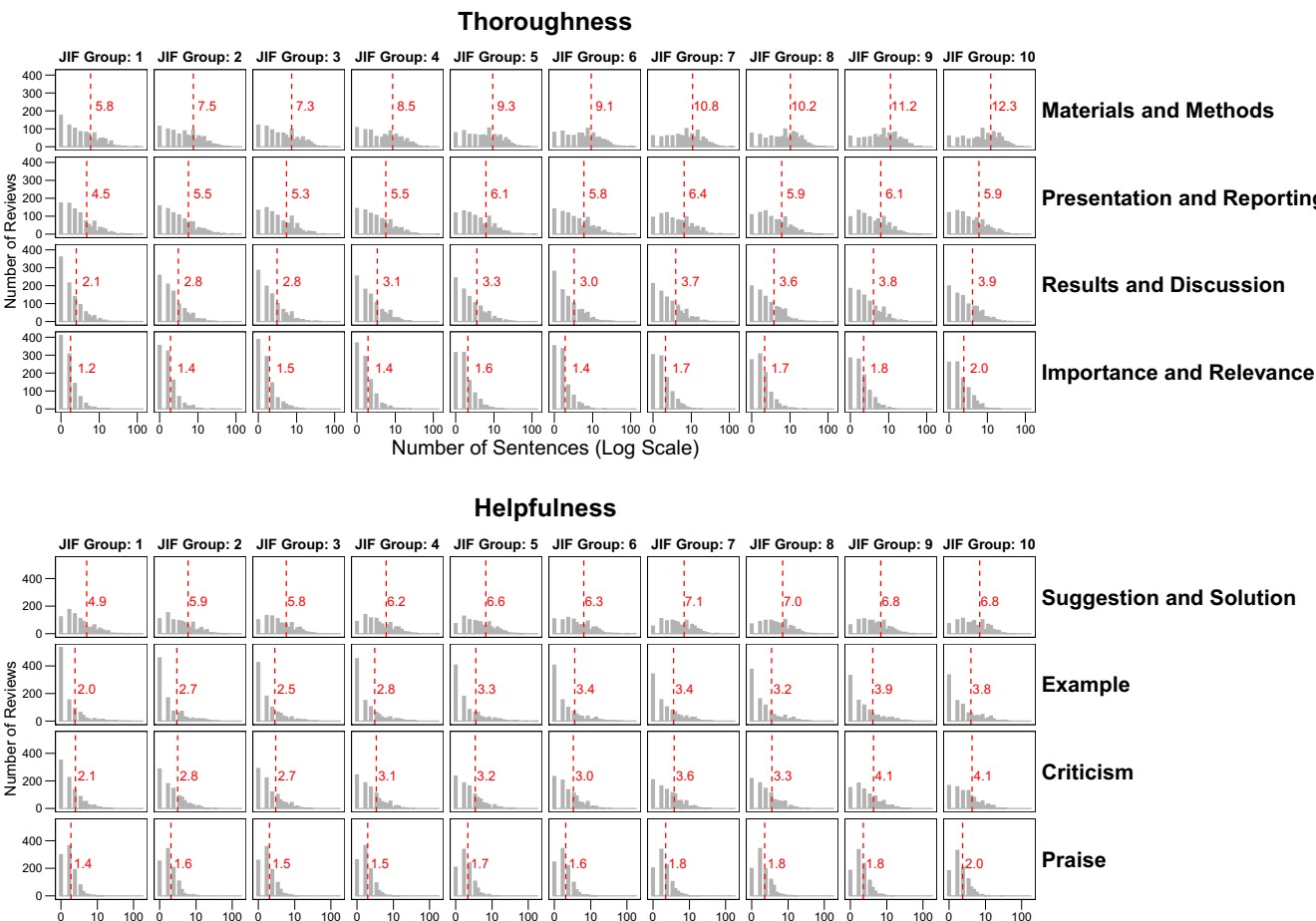

**Fig 2. Distribution of sentences in peer review reports allocated to 8 content categories by *Journal Impact Factor* group.** A sentence could be allocated to no, one, or several categories. Vertical dashed lines show the average number of sentences after aggregating numbers to the level of reviews. The number of sentences are displayed on a log scale. Analysis based on 10,000 review reports. The data underlying this figure can be found in S2 Data.

*Materials and Methods* increased from 40.4% to 51.8% from *Journal Impact Factor* groups 1 to 10. In contrast, attention to *Presentation and Reporting* declined from 32.9% in group 1 to 25.0% in group 10. No clear trends were evident for *Results and Discussion* or *Importance and Relevance*. For helpfulness, the percentage of sentences including *Suggestion and Solution* declined from 36.9% in group 1 to 30.3% in group 10. The prevalence of sentences providing *Examples* increased from 11.0% (group 1) to 13.3% (group 10). *Praise* decreased slightly, whereas *Criticism* increased slightly when moving from group 1 to group 10. The distributions were broad, even within the groups of journals with similar impact factors.

## Regression analyses

The association between journal impact factor and the 8 content categories was analysed in 2 regression analyses. The first predicted the number of sentences of each content category across the 10 *Journal Impact Factor* groups; the second, the changes in the percentage of sentences addressing content categories. All coefficients and standard errors are available from S3 File.

The predicted number of sentences are shown in Fig 4 with their 95% confidence intervals (CI). The results confirm those observed in the descriptive analyses. There was a substantial

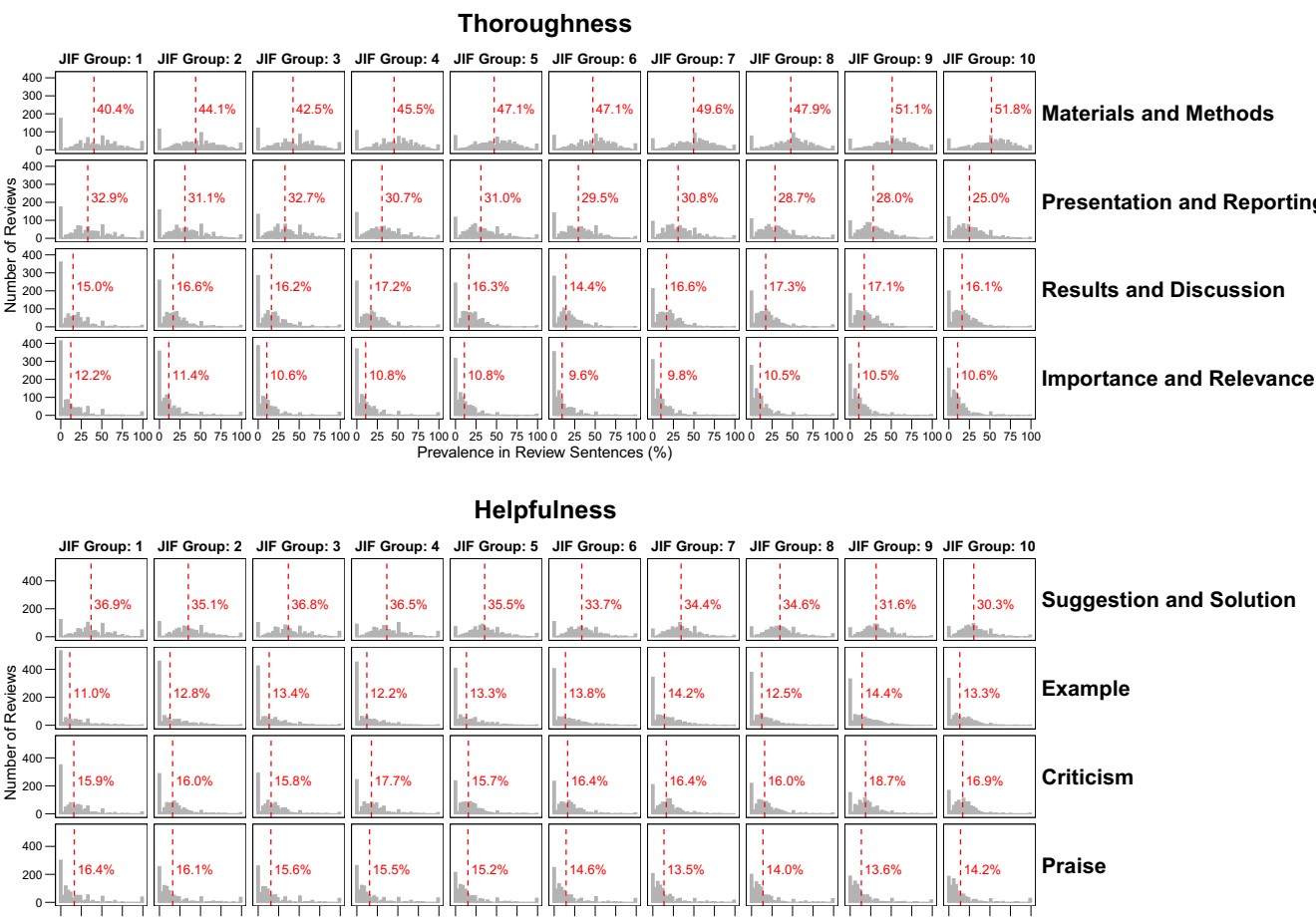

**Fig 3. Distribution of sentences in peer review reports allocated to 8 content categories by *Journal Impact Factor* group.** The percentage of sentences in a review allocated to the 8 peer review quality categories is shown. A sentence could be allocated to no, one, or several categories. Analysis based on 10,000 review reports. Vertical dashed lines show the average prevalence after aggregating prevalences to the level of reviews. The data underlying this figure can be found in S3 Data.

increase in the number of sentences addressing *Materials and Methods* from *Journal Impact Factor* group 1 (6.1 sentences; 95% CI 5.3 to 6.8) to group 10 (12.5 sentences; 95% CI 11.6 to 13.5), for a difference of 6.4 sentences. For the other categories, only small increases were predicted, in line with the descriptive analyses.

The predicted differences in the percentage of sentences addressing content categories are shown in Fig 5. Again, the results confirm those observed in the descriptive analyses. The prevalence of sentences on *Materials and Methods* in the journals with the highest impact factor was higher (+11.0 percentage points; 95% CI + 7.9 to +14.1) than in the group with the lowest impact factor journals. The trend for sentences addressing *Presentation and Reporting* went in the opposite direction, with reviews submitted to the journals with the highest impact factor giving less emphasis to such content (−7.7 percentage points; 95% CI −10.0 to −5.4). There was slightly less focus on *Importance and Relevance* in the group of journals with the highest impact factors relative to the group with the lowest impact factors (−1.9 percentage points; 95% CI −3.5 to −0.4) and little evidence of a difference for *Results and Discussion* (+1.1 percentage points; 95% CI −0.54 to +2.8). Reviews for higher impact factor journals devoted less

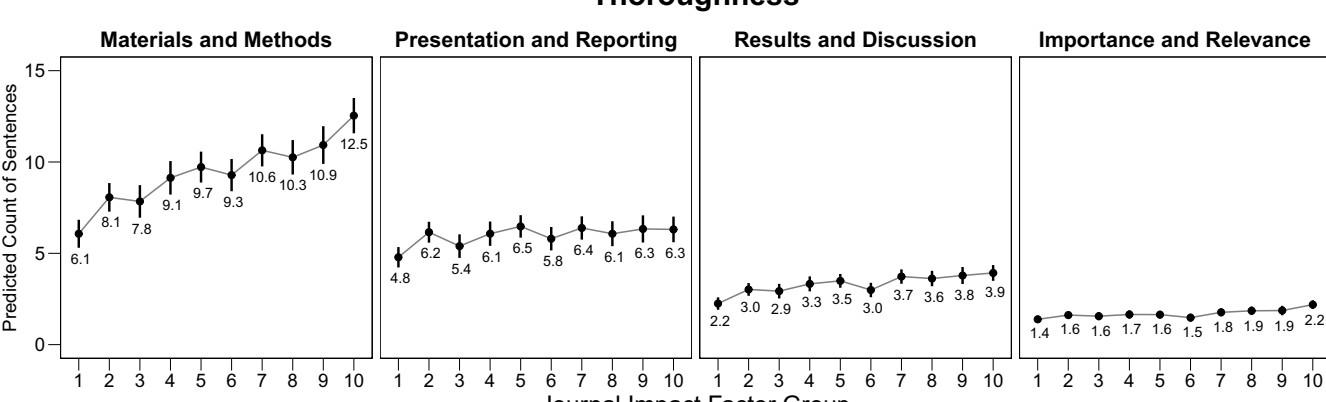

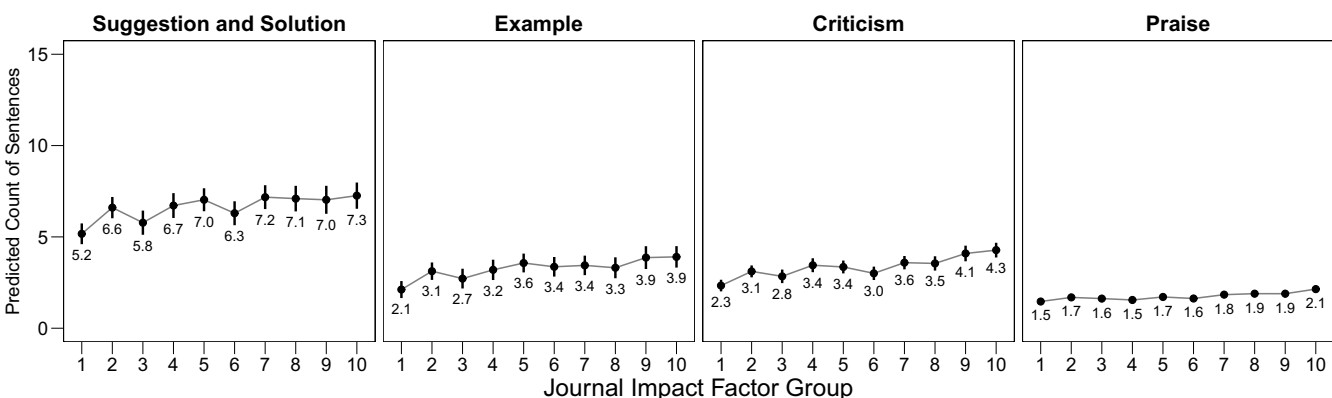

**Fig 4. Predicted number of sentences addressing thoroughness and helpfulness categories across the 10 *Journal Impact Factor* groups.** Predicted values and 95% confidence intervals are shown. Analysis based on 10,000 review reports. All negative binomial mixed-effects models include random intercepts for the journal name and reviewer ID. The data underlying this figure can be found in S4 Data.

attention to *Suggestion and Solution*. The group with the highest *Journal Impact Factor* had 6.2 percentage points fewer sentences addressing *Suggestion and Solution* (95% CI −8.5 to −3.8). No substantive differences were observed for *Examples* (0.3 percentage points; 95% CI −1.7 to +2.3), *Praise* (1.6 percentage points; 95% CI −0.5 to +3.7), and *Criticism* (0.5 percentage points; 95% CI −1.0 to +2.0).

## Sensitivity analyses

We performed several sensitivity analyses to assess the robustness of findings. In the first, we removed reviews with 0 sentences or 0% in the respective content category, resulting in similar regression coefficients and predicted counts. In the second, the sample was limited to reviews with at least 10 sentences (sentence models) or 200 words (percentage model). The analysis showed that short reviews do not drive associations. In the third sensitivity analysis, the regression models adjusted for additional variables (discipline, career stage of reviewers, and log number of reviews submitted by reviewers). The addition of these variables reduced the sample size from 10,000 to 5,806 reviews because of missing reviewer-level data. Again, the relationships between content categories and journal impact factor persisted. The fourth sensitivity

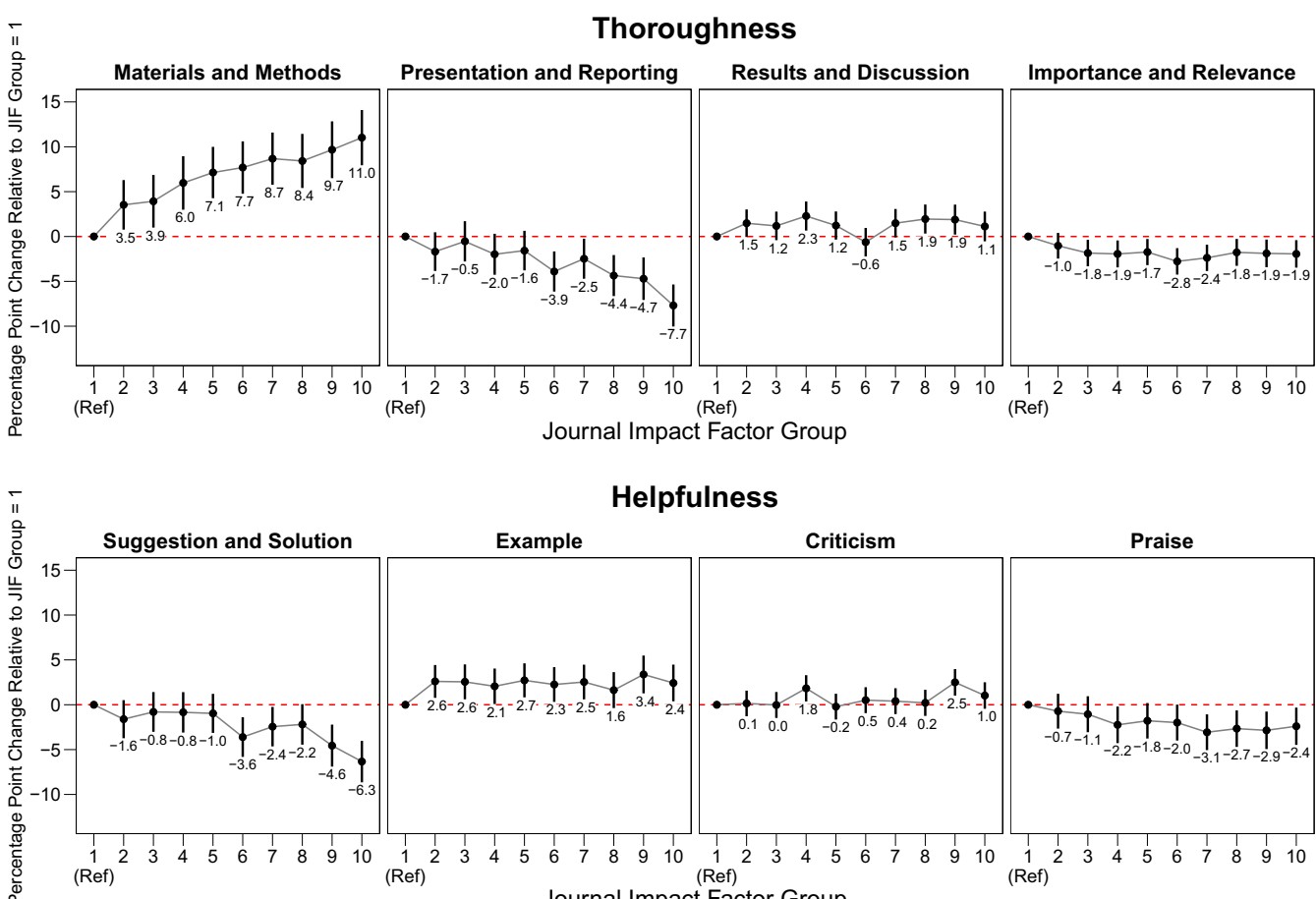

**Fig 5. Percentage point change in the proportion of sentences addressing thoroughness and helpfulness categories relative to the lowest *Journal Impact Factor* group.** Regression coefficients and 95% confidence intervals are shown. Analysis based on 10,000 review reports. All linear mixed-effects models include random intercepts for the journal name and reviewer ID. The data underlying this figure can be found in S5 Data.

analysis revealed that results were generally similar for male and female reviewers. The fifth showed that the results changed little when replacing the *Journal Impact Factor* groups with the raw *Journal Impact Factor* (S3 File).

## Typical words in content categories

A keyness analysis [15] extracts typical words for each content category across the full corpus of the 188,106 sentences. The analysis is based on $\chi^2$ tests comparing the frequencies of each word in sentences assigned to a content category and other sentences. Table 2 reports the 50 words appearing more frequently in sentences assigned to the respective content category than in other sentences (according to the DistilBERT classification). The table supports the validity of the classification. Common terms in the thoroughness categories were "data", "analysis", "method" (*Materials and Methods*); "please", "text", "sentence", "line", "figure" (*Presentation and Reporting*); "results", "discussion", "findings" (*Results and Discussion*); and "interesting", "important", "topic" (*Importance and Relevance*). For helpfulness, common unique words included "please", "need", "include (*Suggestion and Solution*); "line", "page", "figure" (*Examples*); "interesting", "good", "well" (*Praise*); and "however", "(un)clear", "mistakes" (*Criticism*).

**Table 2. The 50 key terms for each content category.** Results rely on keyness analyses using $\chi^2$ tests for each word, comparing the frequency of words in sentences where a content characteristic was present with sentences (target group) where characteristic was absent (reference group). Table reports the 50 words with the highest $\chi^2$ values per category.

| Content category | Words |
|---|---|
| Materials and Methods | data, methods, analysis, model, patients, method, sample, used, analyses, test, treatment, models, performed, using, criteria, control, experiments, statistical, samples, measures, population, group, parameters, measure, approach, methodology, size, measured, procedure, cohort, groups, variables, scale, controls, design, tests, experiment, experimental, selection, testing, tested, measurements, regression, compared, procedures, measurement, analyzed, trials, score, sampling |
| Presentation and Reporting | please, text, sentence, line, figure, written, table, section, page, paragraph, figures, references, introduction, tables, english, abstract, language, word, sentences, description, reference, mention, explain, information, detail, specify, reader, clarify, legend, well, needs, lines, described, mentioned, clearly, describe, term, summarize, details, informative, errors, abbreviations, read, well-written, grammar, explained, remove, check, need, clarified |
| Results and Discussion | results, discussion, findings, conclusions, conclusion, result, outcome, correlation, effect, outcomes, section, finding, interpretation, discussed, correlations, confidence, variance, supported, statistical, regression, significant, implications, discuss, statistically, presented, summarize, main, significance, predictions, analysis, values, deviation, comparison, error, difference, obtained, comparisons, estimates, value, drawn, uncertainty, likelihood, draw, conclude, observed, objective, deviations, discussions, differences, variables |
| Importance and Relevance | interesting, important, topic, interest, research, contribution, field, novel, importance, work, study, audience, relevance, literature, understanding, paper, useful, future, valuable, insights, knowledge, quality, focus, provides, great, originality, overall, rigor, timely, addresses, approach, clinical, significance, relevant, scientific, implications, usefulness, review, general, insight, context, innovative, readership, area, community, revision, comprehensive, findings, perspective, practical |
| Suggestion and Solution | please, need, needs, better, suggest, provide, consider, clarify, recommend, helpful, include, must, section, required, needed, discussion, line, revision, table, detail, remove, discuss, explain, sentence, specify, help, check, revise, text, improve, think, reader, added, delete, make, replace, useful, highlight, minor, comment, might, clarified, details, clearer, paragraph, worth, references, information, adding, perhaps |
| Example | line, page, figure, lines, sentence, paragraph, table, example, replace, delete, legend, remove, please, word, change, line, panel, comma, column, reference, typo, instead, pages, last, page, caption, statement, shown, mean, bottom, sentences, figures, phrase, rephrase, shows, panels, replaced, section, correct, indicate, write, missing, first, figure1, says, confusing, starting, figs, text, meant |
| Criticism | unclear, clear, however, difficult, confusing, don't, missing, hard, lack, sure, lacks, seem, seems, understand, little, misleading, doesn't, enough, vague, confused, incorrect, lacking, unfortunately, somewhat, problematic, insufficient, although, convinced, major, wrong, statement, mistakes, quite, poorly, conclusion, incomplete, questionable, weak, grammatical, inconsistent, errors, sentence, remains, speculative, limited, really, follow, makes, figure, concerns |
| Praise | interesting, well, good, written, well-written, topic, manuscript, paper, important, interest, excellent, overall, satisfactory, comments, timely, nice, great, valuable, author, work, appreciate, review, provides, publication, comprehensive, contribution, article, study, research, novel, useful, enjoyed, field, concise, sound, impressive, improved, dear, easy, nicely, congratulate, thorough, worthy, addresses, relevant, appreciated, appropriate, presents, designed, adequate |

## Discussion

This study used fine-tuned transformer language models to analyse the content of peer review reports and investigate the association of content with the *Journal Impact Factor*. We found that the *impact factor* was associated with the characteristics and content of peer review reports and reviewers. The length of reports increased with increasing *Journal Impact Factor*, with the number of relevant sentences increasing for all content categories, but in particular for

*Materials and Methods*. Expressed as the percentage of sentences addressing a category (and thus standardising for the different lengths of peer review reports), the prevalence of sentences providing suggestions and solutions, examples, or addressing the reporting of the work declined with increasing *Journal Impact Factor*. Finally, the proportion of reviewers from Asia, Africa, and South America also declined, whereas the proportion of reviewers from Europe and North America increased.

The limitations of the *Journal Impact Factor* are well documented [16–18], and there is increasing agreement that it should not be used to evaluate the quality of research published in a journal. The San Francisco Declaration on Research Assessment (DORA) calls for the elimination of any journal-based metrics in funding, appointment, and promotion [19]. DORA is supported by thousands of universities, research institutes and individuals. Our study shows that the peer reviews submitted to journals with higher *Journal Impact Factor* may be more thorough than those submitted to lower impact journals. Should, therefore, the *Journal Impact Factor* be rehabilitated and used as a proxy measure for peer review quality? Similar to the distribution of citations in a journal, the length of reports and the prevalence of content related to thoroughness and helpfulness varied widely, within journals and between journals with similar *Journal Impact Factor*. In other words, the *Journal Impact Factor* is a poor proxy measure for the thoroughness or helpfulness of peer review authors may expect when submitting their manuscripts.

The increase in the length of peer review reports with increasing *Journal Impact Factor* might be explained by the fact that reviewers from Europe and North America and reviewers with English as their first language tend to write longer reports and to review for higher impact journals [20]. Further, high *impact factor* journals may be more prestigious to review for and can thus afford to recruit more senior scholars. Of note, there is evidence suggesting that the quality of reports decreases with age or years of reviewing [21,22]. Interestingly, several medical journals with high *impact factors* have recently committed to improving diversity among their reviewers [23–25]. Unfortunately, due to incomplete data, we could not examine the importance of the level of seniority of reviewers. Independently of seniority, reviewers may be brief reviewing for a journal with low impact factor, believing a more superficial review will suffice. On the other hand, brief reviews are not necessarily superficial: The review of a very poor paper may not warrant a long text.

Peer review reports have been hidden for many years, hampering research on their characteristics. Previous studies were based on smaller, selected samples. An early randomised trial evaluating the effect of blinding reviewers to the authors' identity on the quality of peer review was based on 221 reports submitted to a single journal [26]. Since then, science has become more open, embracing open access to publications and data and open peer review. Some journals now publish peer reviews and authors' responses along with the articles [27–29]. Bibliographic databases have also started to publish reviews [30]. The European Cooperation in Science and Technology (COST) Action on new frontiers of peer review (PEERE), established in 2017 to examine peer review in different areas, was based on data from several hundred Elsevier journals from a wide range of disciplines [31].

To our knowledge, the Publons database is the largest of peer review reports, and the only one not limited to individual publishers or journals, making it a unique resource for research on peer review. Based on 10,000 peer review reports submitted to medical and life science journals, this is likely the largest study of peer review content ever done. It built on a previous analysis of the characteristics of scholars who review for predatory and legitimate journals [32]. Other strengths of this study include the careful classification and validation step, based on the coding by hand of 2,000 sentences by trained coders. The performance of the classifiers was high, which is reassuring given that the sentence-level classification tasks deal with imbalanced and sometimes ambiguous categories. Performance is in line with recent studies. For example,

a study using an extension of BERT to classify concepts such as nationalism, authoritarianism, and trust reported results for precision and recall similar to the present study [33]. We trained the algorithm on journals from many disciplines, which should make it applicable to other fields than medicine and the life sciences. Journals and funders could use our approach to analyse the thoroughness and helpfulness of their peer review. Journals could submit their peer review reports to an independent organisation for analysis. The results could help journals improve peer review, give feedback to peer reviewers, inform the training of peer reviewers, and help readers gauge the quality of the journals in their field. Further, such analyses could inform a reviewer credit system that could be used by funders and research institutions.

Our study has several weaknesses. Reviewers may be more likely to submit their review to Publons if they feel it meets general quality criteria. This could have introduced bias if the selection process into Publons' database depended on the *Journal Impact Factor*. However, the large number of journals within each *Journal Impact Factor* group makes it likely that the patterns observed are real and generalizable. We acknowledge that our findings are more reliable for the more common content categories than for the less common. We only examined peer review reports and could not consider the often extensive contributions made by journal editors and editorial staff to improve articles. In other words, although our results provide valuable insights into the peer review process, they give an incomplete picture of the general quality assurance processes of journals. Due to the lack of information in the database, we could not analyse any differences between open (signed) and anonymous peer review reports. Similarly, we could not distinguish between reviews of original research articles and other article types, for example, narrative review articles. Some journals do not consider importance and relevance when assessing submissions, and these journals may have influenced results for this category. We lacked the resources to identify these journals among the over 1,600 outlets included in our study to examine their influence. Finally, we could not assess to what extent the content of peer review reports affected acceptance or rejection of the paper.

## Conclusions

This study of peer review characteristics indicates that peer review in journals with higher impact factors tends to be more thorough, particularly in addressing the study's methods while giving relatively less emphasis to presentation or suggesting solutions. Our findings may have been influenced by differences in reviewer characteristics, quality of submissions, and the attitude of reviewers towards the journals. Differences were modest, and the *Journal Impact Factor* is therefore a bad predictor of the quality of peer review of an individual manuscript.

## Methods

Our study was based on peer review reports submitted to Publons from January 24, 2014, to May 23, 2022. Publons (part of Web of Science) is a platform for scholars to track their peer review activities and receive recognition for reviewing [34]. A total of 2,000 sentences from peer review reports were hand-coded and assigned to none, one, or more than one of 8 content categories related to thoroughness and helpfulness. The transformer model DistilBERT [14,35] was then used to assign the sentences in peer review reports as contributing or not contributing to categories. More details are provided in the Section "Classification and validation" below and S2 File. After validating the classification performance using out-of-sample predictions, the association between the 2019 *Journal Impact Factors* [36] and the prevalence of relevant sentences in peer review reports was examined. The sample is limited to review reports submitted to medical and life sciences journals with an impact factor. The analysis took the hierarchical nature of the data into account.

## Data sources

As of May 2022, the Publons database contained information on 15 million reviews performed and submitted by more than 1,150,000 scholars for about 55,000 journals and conference proceedings. Reviews can be submitted to Publons in different ways. When scholars review for journals partnering with Publons and wish recognition, Publons receives the review and some meta-data directly from the journal. For other journals, scholars can upload the review and verify it by forwarding the confirmation email from the journal to Publons or by sending a screenshot from the peer review submission system. Publons audits a random subsample of emails and screenshots by contacting editors or journal administrators.

Publons randomly selected English-language peer review reports for the training from a broad spectrum of journals, covering all (ESI) fields [37] except Physics, Space Science, and Mathematics. Reviews from the latter fields contained many mathematical formulae, which were difficult to categorise. In the next step, a stratified random sample of 10,000 verified prepublication reviews written in English was drawn. First, the Publons database was limited to reviews from medical and life sciences journals based on ESI research fields, resulting in a data set of approximately 5.2 million reviews. The ESI field Multidisciplinary was excluded as these journals publish articles not within the medical and life sciences field (e.g., *PLOS ONE*, *Nature*, *Science*). Second, these reviews were divided into 10 equal groups based on *Journal Impact Factor* deciles. Third, 1,000 reviews were selected randomly from each of the 10 groups. Second-round peer review reports were excluded whenever this information was available. The continent of the reviewer's institutional affiliation, the total number of publications of the reviewer, the start and end year of the reviewers' publications, and gender were available for a subset of reviews. The gender of reviewers were classified with the *gender-guesser* Python package (version 0.4.0). Since the data on reviewer characteristics are incomplete and automated gender classification suffers from misclassification, these variables are only included in regression models reported in S3 File.

## Classification and validation

Two authors (ASE and MS) were trained in coding sentences. After piloting and refining coding and establishing intercoder reliability, the reviewers labelled 2,000 sentences (1,000 sentences each). They allocated sentences to none, one, or several of 8 content categories. We selected the 8 categories based on prior work, including the Review Quality Instrument and other scales and checklists [38], and previous studies using text analysis or machine learning to assess student and peer review reports [39–43]. In the manual coding process, the categories were refined, taking into account the ease of operationalising categories and their intercoder reliability. Based on the pilot data, Krippendorff's α, a measure of reliability in content analysis, was calculated [44].

The categories describe, first, the *Thoroughness* of a review, measuring the degree to which a reviewer comments on (1) *Materials and Methods* (Did the reviewer comment on the methods of the manuscript?); (2) *Presentation and Reporting* (Did the reviewer comment on the presentation and reporting of the paper?); (3) *Results and Discussion* (Did the reviewer comment on the results and their interpretation?); and (4) the paper's *Importance and Relevance* (Did the reviewer comment on the importance or relevance of the manuscript?). Second, the *Helpfulness* of a review was examined based on comments on (5) *Suggestion and Solution* (Did the reviewer provide suggestions for improvement or solutions?); (6) *Examples* (Did the reviewer give examples to substantiate his or her comments?); (7) *Praise* (Did the reviewer identify strengths?); and (8) *Criticism* (Did the reviewer identify problems?). Categories were rated on a binary scale (1 for yes, 0 for no). A sentence could be coded as 1 for multiple categories. S4 File gives further details.

We used the transformer model DistilBERT to predict the absence or presence of the 8 characteristics in each sentence of the peer review reports [45]. For validation, data were split randomly into a training set of 1,600 sentences and a held-out test set of 400 sentences. Eight DistilBERT models (one for each content categories) were fine-tuned on the set of 1,600 sentences and predicted the categories in the remaining 400 sentences. Performance measures, including precision (i.e., the positive predictive value), recall (i.e., sensitivity), and the F1 score, were calculated. The F1 score is a harmonic mean of precision and recall and an overall measure of accuracy. The F1 score can range between 0 and 1, with higher values indicating better classification performance [46].

Overall, the classification performance of the fine-tuned DistilBERT language models was high. The average F1 score for the presence of a characteristic was 0.75, ranging from 0.68 (*Praise*) to 0.88 (*Suggestion and Solution*). For most categories, precision and recall were similar, indicating the absence of systematic measurement error. *Importance and Relevance* and *Results and Discussion* were the exceptions, with lower recall for characteristics being present. Balanced accuracy (the arithmetic mean of sensitivity and specificity) was also high, ranging from 0.78 to 0.91 (with a mean of 0.83 across the 8 categories). S2 File gives further details.

We compared the percentages of sentences addressing each category between the human annotation dataset and the output from the machine learning model. For the test set of 400 sentences, the percentage of sentences that fall into each of the 8 categories were calculated, separately for the human codings and the DistilBERT predictions. There was a close match between the two: DistilBERT overestimated *Importance and Relevance* by 3.0 percentage points and underestimated *Materials and Methods* by 2.3 percentage points. For all other content categories, smaller differences were observed. Having assessed the validity of the classification, the machine learning classifiers were fine-tuned using all 2,000 labelled sentences, and the 8 classifiers were used to predict the presence or absence of content in the full text corpus consisting of 188,106 sentences.

Finally, we identified unique words in each quality category using a "keyness" analysis [47]. The words retrieved from the keyness analyses reflect typical words used in each content category.

## Statistical analysis

The association between peer review characteristics and *Journal Impact Factor* groups was examined in 2 ways. The analysis of the number of sentences for each category used negative binomial regression models. The analysis of the percentages of sentences addressing content categories relied on linear mixed-effects models. To account for the clustered nature of the data, we include random intercepts for journals and reviewers [48]. The regression models take the form,

$$Y_i = \alpha_{j[i],k[i]} + \sum_{m=2}^{10} \beta_m \cdot \mathbb{I}(JIF_i = m) + \epsilon_i$$

with

$$\alpha_j \sim N\left(\mu_{\alpha_j}, \sigma^2_{\alpha_j}\right), \text{ for journal } j = 1, \ldots, J.$$

$$\alpha_k \sim N\left(\mu_{\alpha_k}, \sigma^2_{\alpha_k}\right), \text{ for reviewer } k = 1, \ldots, K$$

$$\epsilon_i \sim N(0, \sigma^2)$$

where $Y_i$ is the count of sentences addressing a content category (for the negative binomial regression models) or the percentages (for the linear-mixed effects models), while $i$, $\beta_m$ are the coefficients for the $m = 2, . . ., 10$ categories of the categorical variable of *Journal Impact Factor* (with $m = 1$ as the reference category), and $\epsilon_i$ is the unobserved error term. The model includes varying intercepts $\alpha_{j[i],k[i]}$ for $J$ journals and $K$ reviewers. $\mathbb{I}(\cdot)$ denotes the indicator function.

All regression analyses were done in R (version 4.2.1). The fine-tuning of the classifier and sentence-level predictions were done in Python (version 3.8.13). The libraries used for data preparation, text analysis, supervised classification, and regression models were *transformers* (version 4.20.1) [49], *quanteda* (version 3.2.3) and *quanteda.textstats* (version 0.95) [50], *lme4* (version 1.1.30) [51], *glmmTMB* (version 1.1.7) [52], *ggeffects* (version 1.1.5) [53], and *tidyverse* (version 1.3.2) [54].

## Supporting information

**S1 File. Journals and disciplines included in the study.** The 10 journals from each journal impact factor group that provided the largest number of peer review reports and all 1,664 journals included in the analysis listed in alphabetical order. The numbers in parentheses represent the JIF and the number of reviews included in the sample.
(PDF)

**S2 File. Further details on classification and validation.** Further information on the hand-coded set of sentences, the classification approach, and performance provide metrics on the classification performance and show that aggregating the classification closely mirrors human coding of the same set of sentences. All results are out-of-sample predictions, meaning that the data in the held-out test set are not used for training the classifier during validation steps.
(PDF)

**S3 File. Additional details on regression analyses and sensitivity analyses.** All regression tables for the analysis reported in the paper, and plots and regression tables relating to the 5 sensitivity analyses. All sensitivity analyses are conducted for the prevalence-based and sentence-based models.
(PDF)

**S4 File. Codebook and instructions.** Coding instructions and examples for each of the 8 characteristics of peer review reports.
(PDF)

**S1 Data. Supporting data for Fig 1.**
(XLSX)

**S2 Data. Supporting data for Fig 2.**
(XLSX)

**S3 Data. Supporting data for Fig 3.**
(XLSX)

**S4 Data. Supporting data for Fig 4.**
(XLSX)

**S5 Data. Supporting data for Fig 5.**
(XLSX)

**S6 Data. Supporting data for Fig 1 in S1 File.**
(XLSX)

**S7 Data. Supporting data for Fig 1 in S2 File.**
(XLSX)

**S8 Data. Supporting data for Fig 2 in S2 File.**
(XLSX)

**S9 Data. Supporting data for Fig 3 in S2 File.**
(XLSX)

**S10 Data. Supporting data for Fig 4 in S2 File.**
(XLSX)

**S11 Data. Supporting data for Fig 1 in S3 File.**
(XLSX)

**S12 Data. Supporting data for Fig 2 in S3 File.**
(XLSX)

**S13 Data. Supporting data for Fig 3 in S3 File.**
(XLSX)

**S14 Data. Supporting data for Fig 4 in S3 File.**
(XLSX)

**S15 Data. Supporting data for Fig 5 in S3 File.**
(XLSX)

**S16 Data. Supporting data for Fig 6 in S3 File.**
(XLSX)

**S17 Data. Supporting data for Fig 7 in S3 File.**
(XLSX)

**S18 Data. Supporting data for Fig 8 in S3 File.**
(XLSX)

**S19 Data. Supporting data for Fig 9 in S3 File.**
(XLSX)

**S20 Data. Supporting data for Fig 10 in S3 File.**
(XLSX)

## Acknowledgments

We are grateful to Anne Jorstad and Gabriel Okasa from the Swiss National Science Foundation (SNSF) data team for valuable comments on an earlier draft of this paper. We would also like to thank Marc Domingo (Publons, part of Web of Science) for help with the sampling procedure.

## Author Contributions

**Conceptualization:** Anna Severin, Matthias Egger, Tiago Barros, Stefan Müller.

**Data curation:** Michaela Strinzel, Matthias Egger, Alexander Sokolov, Stefan Müller.

**Formal analysis:** Alexander Sokolov, Stefan Müller.

**Funding acquisition:** Matthias Egger.

**Investigation:** Anna Severin, Michaela Strinzel, Matthias Egger, Tiago Barros, Alexander Sokolov, Julia Vilstrup Mouatt, Stefan Müller.

**Methodology:** Anna Severin, Matthias Egger, Tiago Barros, Julia Vilstrup Mouatt, Stefan Müller.

**Project administration:** Anna Severin, Michaela Strinzel, Matthias Egger.

**Resources:** Anna Severin, Matthias Egger, Stefan Müller.

**Software:** Stefan Müller.

**Supervision:** Matthias Egger, Julia Vilstrup Mouatt.

**Validation:** Anna Severin.

**Writing – original draft:** Anna Severin.

**Writing – review & editing:** Anna Severin, Michaela Strinzel, Matthias Egger, Tiago Barros, Alexander Sokolov, Julia Vilstrup Mouatt, Stefan Müller.

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
