## [Editor Report · Decision Letter 0]

23 Aug 2022

Dear Matthias, 

Thank you for submitting your manuscript entitled "Journal Impact Factor and Peer Review Thoroughness and Helpfulness: A Supervised Machine Learning Study" for consideration as a Meta-Research Article by PLOS Biology.

Your manuscript has now been evaluated by the PLOS Biology editorial staff, as well as by an academic editor with relevant expertise, and I'm writing to let you know that we would like to send your submission out for external peer review.

Once your full submission is complete, your paper will undergo a series of checks in preparation for peer review. After your manuscript has passed the checks it will be sent out for review. To provide the metadata for your submission, please Login to Editorial Manager (https://www.editorialmanager.com/pbiology) within two working days, i.e. by Aug 25 2022 11:59PM.

Kind regards,

Roli

Roland Roberts, PhD

Senior Editor

PLOS Biology

rroberts@plos.org

---

## [Decision Letter · Decision Letter 1]

12 Oct 2022

Dear Matthias,

Thank you for your patience while your manuscript "Journal Impact Factor and Peer Review Thoroughness and Helpfulness: A Supervised Machine Learning Study" was peer-reviewed at PLOS Biology. Your manuscript has been evaluated by the PLOS Biology editors, an Academic Editor with relevant expertise, and by three independent reviewers.

As you will see in the reviewer reports, which can be found at the end of this email, although the reviewers find the work potentially interesting, they have also raised a substantial number of important concerns. Based on their specific comments and following discussion with the Academic Editor, it is clear that a substantial amount of work would be required to meet the criteria for publication in PLOS Biology.

However, given our and the reviewers' interest in your study, we would be open to inviting a comprehensive revision of the study that thoroughly addresses all the reviewers' comments. Given the extent of revision that would be needed, we cannot make a decision about publication until we have seen the revised manuscript and your response to the reviewers' comments. Your revised manuscript would need to be seen by the reviewers again, but please note that we would not engage them unless their main concerns have been addressed. 

You'll see that most of reviewer #1's requests are for methodological clarification, but he also wonders whether your results may be skewed by the greater length of the reviews in high-impact journals. He also points out the underappreciated contribution of professional editors (!!). Reviewer #2 is also broadly positive, though regrets the fact that data are from a single source; he recommends that you try to validate with respect to another source, and suggests several substantial additional analyses. Like reviewer #1, he requests some methodological clarifications, mentions some possible confounders, and makes some very helpful textual and interpretational suggestions. Reviewer #3, who is a machine learning expert, is similarly positive, but has a long series of quite severe-sounding criticisms of your methodology and its reporting.

We appreciate that these requests represent a great deal of extra work, and we are willing to relax our standard revision time to allow you 6 months to revise your study. Please email us (plosbiology@plos.org) if you have any questions or concerns, or envision needing a (short) extension.

**IMPORTANT - SUBMITTING YOUR REVISION**

*Resubmission Checklist*

*Published Peer Review*

*PLOS Data Policy*

Sincerely,

Roli

Roland Roberts, PhD

Senior Editor

PLOS Biology

rroberts@plos.org

REVIEWERS' COMMENTS:

Reviewer #1:

[identifies himself as Ludo Waltman]

Please find my review at https://ludowaltman.pubpub.org/pub/review-jif-pr/release/1

[the editor has here pasted the text of the review from that location]

This paper presents a large-scale analysis of the content of peer review reports, focusing on different types of comments provided in review reports and the association with the impact factors of journals. The scale of the analysis is impressive. Studies of the content of such a large number of review reports are exceptional. I enjoyed reading the paper, even though I did not find the results presented in the paper to be particularly surprising.

Feedback and suggestions for improvements are provided below.

The methods used by the authors would benefit from a significantly more detailed explanation:

“Scholars can submit their reviews for other journals by either forwarding the review confirmation emails from the journals to Publons or by sending a screenshot of the review from the peer review submission system.”: This sentence is unclear. Review confirmation emails often do not include the review itself, only a brief ‘thank you’ message, so it is not clear to me how a review can be obtained from such a confirmation email. I also do not understand how a review can be obtained from a screenshot. A screenshot may show only part of the review, not the entire review, and there would be a significant technical challenge in converting the screenshot, which is an image, to machine-readable text.

I would like to know whether all reviews are in English or whether there are also reviews in other languages.

Impact factors change over time. New impact factors are calculated each year. The authors need to explain which impact factors they used.

There are many journals that do not have an impact factor. The authors need to explain how these journals were handled.

The authors also need to discuss how reviewers were linked to publication profiles. This is a non-trivial step that needs to be taken to determine the number of publications of a reviewer and the start and end year of the publications of a reviewer. The authors do not explain how this step was taken in their analysis. It is important to provide this information.

“We used a Naïve Bayes algorithm to train the classifier and predict the absence or presence of the eight characteristics in each sentence of the peer review report.”: The machine learning approach used by the authors is explained in just one sentence. A more elaborate explanation is needed. There are lots of machine learning approaches. The authors need to explain why they use Naïve Bayes. They also need to briefly discuss how Naïve Bayes performs the classification task.

Likewise, I would like to see a proper discussion of the statistical model used by the authors. The authors informally explain their statistical approach. I would find it helpful to see a more formal description (in mathematical notation) of the statistical model used by the authors.

“Most distributions were skewed right, with a peak at 0% showing the number of reviews that did not address the content category (Fig 1).”: I do not understand how the peaks at 0% can be explained. Could this be due to problems in the data (e.g., missing or empty review reports)? The authors need to explain this.

“the prevalence of content related to thoroughness and helpfulness varied widely even between journals with similar journal impact factor”: I am not sure whether the word ‘between’ is correct in this sentence. My understanding is that the authors did not distinguish between variation between journals and variation within journals.

“Some journals now publish peer reviews and authors' responses with the articles”: Consider citing the following paper: https://doi.org/10.1007/s11192-020-03488-4. I also recently published a blog post on this topic: https://www.leidenmadtrics.nl/articles/the-growth-of-open-peer-review.

“Bibliographic databases have also started to publish reviews.”: In addition to Web of Science, I think the work done by Europe PMC needs to be acknowledged as well. See for instance this poster presented at the recent OASPA conference: https://oaspa.org/wp-content/uploads/2022/09/Melissa-Harrison_COASP-2022-poster_V2.pdf.

“peer review in journals with higher impact factors tends to be more thorough in addressing study methods but less helpful in suggesting solutions or providing examples”: I wonder whether this conclusion is justified. Relatively speaking sentences in reviews for higher impact factor journals are indeed more likely to address methods and less likely to suggest solutions or to provide examples. However, as shown by the authors, reviews for higher impact factor journals tend to be substantially longer than reviews for lower impact factor journals. Therefore it seems that the total number of sentences (as opposed to the proportion of sentences) suggesting solutions or providing examples may be higher in reviews for higher impact factor journals than in reviews for lower impact factor journals. If that is indeed the case, it seems to me the conclusion should be that peer review in higher impact factor journals is both more thorough and more helpful.

Finally, I think it needs to be acknowledged that quality assurance processes of journals consist not only of the work done by peer reviewers but also of the work done the editorial staff of journals. This seems important in particular for more prestigious journals, which presumably make more significant investments in editorial quality assurance processes. The results presented in the paper offer valuable insights into peer review processes, but they provide only a partial picture of the overall quality assurance processes of journals.

Reviewer #2:

[identifies himself as Bernd Pulverer]

Severin et al. add to their previous work (ref 23) on analyzing attributes of scholarly referee reports. Peer review is generally regarded as a pivotal component of the scholarly process and as such quantitative analysis is to be welcomed. 

The study is based on a large set of about 10,000 referee report across a broad set if biomedical disciplines and uses human annotation to train machine learning based extraction according to 8 pre-identified categories. The study limits itself to analyzing how the 8 referee report attributes compare across 10 Journal Impact Factor (JIF) bins. Regression modelling is applied. Several of the attributes exhibit no trends, others at best very weak trends. That in itself is notable, as for example the minor negative trend of comments on 'Importance and Relevance' vs. JIF is surprising as higher JIF journals tend to instruct referees to comment specifically on these attributes, which for a core part of the selection criteria of such journals. Stronger correlations are reported for the categories 'Materials and Methods' (positive), 'Presentation and Reporting' and 'Suggestion and Solution' (both negative). The authors conclude that referee reports for higher JIF journals may be more 'thorough' but less 'helpful in suggesting solutions and providing examples'. These trends are notable and not predictable - they are also somewhat difficult to rationalize and it is to the authors credit that they don't overinterpret these numbers beyond the conclusions that 'JIF is a bad predictor of peer review' and in fact end the paper with a balanced strength/weakness analysis. The data and approach as reported in detail, but source data for fig 1-3 should be added.

This is an important area of analysis of general interest and the study is thorough. The conclusions are somewhat limited by restricting the analysis to one variable, the JIF, and by limiting the referee report attributes to 8 categories (see below). With the heavy lifting of the human curated training set in hand, it is a pity that the study was not developed beyond the JIF correlations. As such, this specific analysis appears novel and it is based on a large dataset, albeit form a single source.

Major comments:

1) The dataset is large, but limited to one database (Publons). This may well add biases the data, as the authors note themselves. It would have been helpful to expand the analysis to other databases hosting referee reports, such as ORCID, as well as to journals that publish referee reports alongside their papers, such as BMJ, EMBO, eLife and some Nature branded journals. Minimally to test if the reported trends hold up. 

2) It would also have been useful to test for another potential bias: open reports vs. closed reports (still the majority): a collaboration with journals that do not publish their reports (and filtering out referees who posted on Publons or ORCID) would have led to an interesting comparison if the trends are identical when peer review is confidential. Since only aggregate data are reported a journal/publisher collaboration should be feasible. 

3) The study is based on 8 categories. It is unclear how these categories were chosen and the detail of how the annotators defined them is limited. More importantly, other important attributes are missing, for example number of experimental requests made vs. number of textual requests made, or % of a referee report dedicated to specific points vs. general discussion/subjective points. Expanding the set would add value. As a minor point, it is noted that one category, 'Importance and Relevance' is explicitly excluded from a number of major journals, such as PLOSONE. This could be a confounding factor. I realize that 'multidisciplinary' journals have already been excluded, and maybe this covers all such journals, but please comment.

4) It is unclear if only research papers were analyzed. It is recommended that other peer reviewed papers such as reviews are excluded.

5) The study shows that the JIF does not predict many of the attributes. With the same dataset other variables could be assessed, such as 'subject area' (already defined in the study as ESI research field, in particular clinical research vs. 'basic research'). This is in particular important as baseline JIF is rather different between such categories, which may be a confounding factor in this analysis, but it may also lead to stronger correlations than JIF. Other variables could be category of paper (short report vs. full research paper) or length of paper (report correlation between paper and referee report length). Other interesting areas would be journal name, journal editorial process, referee age or experience, referee gender, referee affiliation. The authors note that referee age could not be analyzed and discuss other variables noting 'adjusting for additional variables strengthened relationships'. It is recommended that this section is expanded and the data added. The referee geography as a function of JIF is reported in Table 1: it would be interesting to correlate this with that of the corresponding authors of the paper refereed, if that is feasible.

6) The 'trends' seen for 'Importance and Relevance' and 'Example' (fig 3) are reported as statistically significant, but they are very small and arguable hard to interpret on the background of complex confounding factors. 'Criticism' shows arguable a similar range of variation and yet is classed as 'no effect'. I would recommend not to emphasize these.

Minor Comments:

1) The very first sentence of the abstract states that JIF is used as a proxy for journal quality and thus peer review quality. First of all, JIF claims to measure 'impact' not quality and this is a distortion, although both may correlate. Also, as stated it is implied that referees select what is published, which is not the case (editors select assisted by referee input). Thus, even assuming editors select for JIF maximization, the JIF to peer review connection is indirect at best.

2) Please explain why the regression analyses were controlled for review length. 'since longer texts …address more categories' seems tenuous as multiple categories can be assigned to each sentence.

3) Discussion, second paragraph: a key outcome of studies such as this is to develop 'referee credit' systems. Processes such as the referee report analysis applied here can be applied to individual referee reports and referees to aid such a system.

4) The 'Typical words' section could be removed as it is covered in S4.

5) Table 2 is of limited value and could be removed or added as a supplementary figure.

Text suggestions:

1) Abstract , line 3: add also no. papers analyzed here.

2) Abstract, line 9: state whole range to avoid confusion: 0.21-74.70, median 1.2-8.0

3) Introduction, lines 4-9: I suggest to remove the claim that peer review is 'particularly critical for the medical sciences'. This is debatable, but the paper is not restricted to the medical sciences (in fact, as noted above, a comparison between medical and biological sciences in this dataset would add considerably).

4) Introduction, second para: the claim 'in the absence of evidence on the quality of peer review ….proxy measures like JIF…' is tenuous at best. JIF is used as a proxy for 'impact' maybe even 'quality' and peer review is a key part of quality assurance but does not in itself define journal selection. In fact, one could highlight two functions: aiding journal selection; improving paper. Please adapt.

5) Introduction, second para: change 'articles published' to 'articles classed as citable (by ISI-Clarivate)'

6) Discussion, line 14: I am not sure the data definitively show high JIF reports are 'more thorough'.

7) Discussion, 3rd para: I disagree with the hypothesis that 'junior referees might be less able to comment on methodology'. All the evidence point the other way, and this is not surprising since ECRs are practitioners. It is fine to pose a hypothesis of course and then to cite evidence again, but this section could also be deleted as it is - unfortunately - not tested here.

8) Discussion 4th para: Ref 19, 20 are cited in support of transparent/open peer review. Nature was actually rather late in adopting this and others, like BMJ group, EmboPress, BMC series and eLife, could be cited.

9) Discussion 5th para: ORCID should be discussed here.

10) Methods: Publons has been part of Clarivate for years not 'now'.

This referee I not an expert in machine learning or statistical analysis and did therefore not assess these aspects of the work in detail.

Reviewer #3:

While this is a highly interesting study, there are several major questions and issues that preclude a favorable assessment at this point.

1) Introduction

It´s relatively well known, at least in my circles, that the impact factor (IF) is misused to assess journal quality and even single paper quality. However, I do not necessarily agree with the notion that this included an overestimation of peer review quality as well. Curiously, the manuscript also does not provide a single reference to back this extension ("re used to assess the quality of journals and, by extension, the quality of peer review."). This is problematic since the premise of the introduction lies upon this idea. 

The authors should provide evidence for the notion that IF and peer review quality are linked or have been perceived as linked. 

2) Methods

General comment on the methods: The machine learning pipeline is not very well described, also with the added supplement. A supplement should contain no information that is absolutely necessary to understand the methodology.

I strongly suggest a general revision for clarity using standard machine learning terminology and phrasings, and review what is in the main manuscript and the supplement. 

One example:

"We divided the sample into five equally sized subsets and ran the cross-validation five times."

This is how cross-validation was explained. While this explanation _could_ in theory mean cross-validation, this definition could also support other data splits which are not cross-validation. 

3) Methods / Classification and Validation

In the section that describes the categories, I noted that some points were labeled as "did the reviewers _discuss_" a certain topic, whereas in others the label was "did the reviewers _comment on_" a topic. 

To comment on something or to discuss something is a clear qualitative difference. Is there a reason why these phrasings were used? 

4) Methods / metrics

The method sections reads as if the authors only calculated PPV, sensitivity, and the F1 score. However, the supplement S2 also describes and shows the accuracy. I assume that the authors calculated an even bigger set of metrics. So please justify why these particular 3 (or 4) metrics were chosen to be presented in the paper (and why others were not). 

5) Methods / Results

Generally, the performance of the models is pretty bad. Even the top three that were chosen for further analysis perform pretty badly. Also, the authors have compared only NB and an SVM. If this was a data science project in a bootcamp, the authors would fail it as they kind of stopped after 40% of the work. Especially boosted trees would have been worth exploring as they consistently rank highest in the literature and in competition compared to simpler algorithms. Together with point 6 (below) and proper hyperparameter tuning it is likely that a boosted tree model would lead to better results. 

6) Methods 

Looking at S2, it becomes pretty clear that there is an imbalance problem, definitely present for the 5 least common categories. I did not find any mention that the authors adjusted their k-fold crossvalidation for imbalanced data. In this case stratified sampling for the k-fold cross validation is the right method which would likely lead to better and more stable results. To assess this the authors should also report the standard deviation of the cross-validation. I would generally also suggest 10 instead of 5 folds when dealing with such a more complicated setup. 

7) Results

How the paper is written, it is very suggestive that the authors believe that the correlation found between high-impact journals and peer-review focusing more on methods is also causal. While the authors to not claim causality (that per definition cannot be shown by ML alone), the phrasings are still very suggestive, e.g. from the abstract: "In conclusion, peer review in journals with higher journal impact factors tends to be more thorough in discussing the methods used...". There is, however, also the mention of a confounding factor. The authors say that reviewers for high impact journals tended to come from a certain geographic (Europe/NA). So, maybe researchers in Europe/NA are trained to be more focused on methods? Given this confounder and the fact that ML is based on correlation, any conclusions drawn from this study should be phrased much more carefully than currently. 

8) General comment

Given how the study was based on available peer reviews the categories for very high impact of course contain those high-impact journals but very well known highest-impact journals like new england journal of medicine are not in the top ten. Are such flagship journals even present in the sample? If not that´s a shortcoming and a limitation. The authors should comment on this.

Overall, the methodological shortcomings of the study make it hard for me find the results trustworthy. The ML modelling should be performed completely and according to state-of-the-art and conclusions should only be drawn on data generated by those final models.

---

## [Decision Letter · Decision Letter 2]

9 May 2023

Dear Dr Egger,

Thank you for your patience while we considered your revised manuscript "Journal Impact Factor and Peer Review Thoroughness and Helpfulness: A Supervised Machine Learning Study" for consideration as a Meta-Research Article at PLOS Biology. Your revised study has now been evaluated by the PLOS Biology editors, the Academic Editor, and the original reviewers. 

In light of the reviews, which you will find at the end of this email, we are pleased to offer you the opportunity to address the [comments/remaining points] from the reviewers in a revision that we anticipate should not take you very long. We will then assess your revised manuscript and your response to the reviewers' comments with our Academic Editor aiming to avoid further rounds of peer-review, although might need to consult with the reviewers, depending on the nature of the revisions.

IMPORTANT - Please attend to the following:

a) Reviewer #1 raises a potentially important point about your decision to normalise by length, and indeed all three reviewers mention the way that review length was treated in this study. Please address these and the other concerns raised by the reviewers.

b) Please could you change the Title to "Relationship between Journal Impact Factor and the thoroughness and helpfulness of peer reviews"? Normally we would ask you to incorporate the specific finding(s) in the title, but these are somewhat complex and nuanced, and may change in response to the reviewers' comments.

c) Please ensure that you comply with our Data Policy; specifically, we need you to supply the numerical values underlying Figs 1, 2, 3, S1.1, S2.1, S2.2, S2.3, S2.4, and the 4 Figs in “S3 File”, either as a supplementary data file or as a permanent DOI’d deposition. We note that you have plans to deposit the data and R code in the Harvard Metaverse; however, I will need to see this before accepting the paper for publication, and we will also need you to make a permanent DOI’d version (e.g. in Zenodo).

d) Please cite the location of the data clearly in all relevant main and supplementary Figure legends, e.g. “The data underlying this Figure can be found in S1 Data” or “The data underlying this Figure can be found in https://doi.org/10.5281/zenodo.XXXXX”

**IMPORTANT - SUBMITTING YOUR REVISION**

*Resubmission Checklist*

*Published Peer Review*

*PLOS Data Policy*

*Blot and Gel Data Policy*

Sincerely,

Roli Roberts

Roland Roberts, PhD

Senior Editor

PLOS Biology

rroberts@plos.org

REVIEWERS' COMMENTS:

Reviewer #1:

[identifies himself as Ludo Waltman]

I am pleased to see the significant improvements made by the authors to their paper. I have one remaining comment.

The authors conclude that "this study of peer review characteristics indicates that peer review in journals with higher impact factors tends to be more thorough in addressing study methods but less helpful in suggesting solutions or providing examples". As pointed out in my previous review, I don't think this conclusion is warranted. It disregards the fact that review reports in journals with higher impact factors are much longer, on average, than review reports in journals with lower impact factors. The percentage of sentences in review reports that suggest solutions or provide examples is lower for higher impact factor journals than for lower impact factor journals, but the absolute number of sentences suggesting solutions or providing examples is higher, not lower. In my view, the conclusion therefore should be that journals with higher impact factors provide reviews that are more, not less, helpful in suggesting solutions or providing examples.

In their response to my previous report, the authors point out that "our analyses controlled for the length of peer review". This is exactly the problem. All statistics presented in the paper are percentages rather than absolute numbers, so the authors indeed control for the length of a review report. However, length is a relevant factor that, I would argue, one should not necessarily control for. For instance, suppose we have two review reports. One has a length of 100 words, 50 of which are used to provide suggestions. The other has a length of 1000 words, 200 of which are used to provide suggestions. From a relative point of view, the former report is more helpful in providing suggestions (50% vs. 20% of the words are used to provide suggestions), but from an absolute point of view, the latter report is more helpful in providing suggestions (50 vs. 200 words are used to provide suggestions). In my view, the absolute perspective is more relevant. The latter report is the one that will be more helpful for authors to improve their work.

More generally, the fact that review reports in the highest impact factor category are more than twice as long as review reports in the lowest impact factor category is of major importance and, in my view, needs to be emphasized more strongly. It indicates that higher impact factor journals tend to offer more in-depth peer review than lower impact factor journals. This is an important finding that I believe should be mentioned In the abstract and in the concluding section.

Ludo Waltman

PS I published my previous review online. I had hoped to also publish my new review. However, it seems the authors haven't posted their revised paper on a preprint server. I therefore consider the revised paper to be confidential and I won't publish my review.

Reviewer #2:

[identifies himself as Bernd Pulverer]

Ref #2 Re-review:

The authors are to be commended for the thorough responses.

A number of comments:

1) I appreciate the point that many journals with open review processes share these on Publons (now part of Clarivate 'Web of science'). However, Journal with public but unsigned reports less so. Nonetheless, I agree that scraping the literature for non-Publons listed reports in the absence of standardize identifiers is not trivial. I do believe ORCID profiles can point to referee reports from the 'Review URL' field (cf. https://support.orcid.org/hc/en-us/articles/360006971333-Peer-Review). This study us based on a large dataset and it is certainly reasonable to restrict the study to this dataset as it is unclear if a broader set of input data would alter the conclusions significantly. These points could be discussed.

2) Thank you for pointing out that signed vs. unsigned reports and referee reports on research article vs. reviews was not assessed - that is fine, but I am unclear why signed reports could not be automatically identified and compared to unsigned reports. Note that I had suggested a third comparison between published and unpublished referee reports, but acknowledge that while very interesting, this would be a complex undertaking that can be discussed. 

3) - 6) Thank you for the constructive comments and revision

The minor points are addressed, apart from point 3): I would recommend to highlight more clearly that automated analysis of referee reports for quality attributes may inform a referee credit system that could be used objectively and at scale in research assessment by funders and research institutions.

I assessed the responses to ref #1 and #3 and, leaving aside the technical details on statistics and models, which I did not judge, I believe the responses are thorough and the revisions comprehensive leading to a more informative and balanced manuscript. In particular the causality point by ref 3 (no 7) is important and was addressed.

It may be worth emphasizing the referee report length more, both as a correlation with JIF and in the context of the length control applied here (as discussed in ref #1, point 10; ref #2, minor point2).

Reviewer #3:

The re-work of the manuscript was extensive, the authors have addressed all relevant shortcomings very well. 

I have only two minor comments, that should be addressed imo before publication. 

1) discussion, p. 12

"Our study shows that the peer reviews submitted to journals with higher Journal Impact Factor may be more thorough than those submitted to lower impact journals. Should, therefore, the Journal Impact Factor be rehabilitated, and used as a proxy measure for peer review quality? "

In the following discussion of this question, and also at other parts of the manuscript, there is the implicit assumption that submitted peer reviews are independent of the impact factor and journal, i.e. the same effort is put into providing a review. But of course that is likely not true. That reviews tend to be shorter and less thorough w/r to methodology in lower impact journals can have two additional confounding factors:

a) People are less thorough _because_ it is a low impact journal, believing they do not need to provide a review of as good quality as for a journal with a higher impact factor. 

b) People might also be less thorough, when only basing this on the length(!), because the quality does not warrant more text. Let me explain. If I am confronted with an applied AI in healthcare methodology, that is completely not up to the standards, I might just write exactly that and give some examples in bullet points, and suggest rejection. Confronted with a good methodology that has only _some_ major shortcomings, I will likely take the time (and words) to explain these few shortcomings. My point here is that longer -> "more thorough" does not necessary mean more useful or better. The former case does not _warrant_ more text (and it also does not warrant many suggestions. Is it true that I find the former more often in low impact journals? I do not know. I had one of my worst experiences in this regard in the flagship journal of my field (and the paper was accepted despite the fact that I suggested a reject as the only AI methodology expert). But I think this point could still be considered. 

I believe that the discussion would improve if these additional points were also discussed as potentially confounding factors and why this topic is very hard to assess. 

2) conclusion, p. 14

"This study of peer review characteristics indicates that peer review in journals with higher impact factors tends to be more thorough in addressing study methods but less helpful in suggesting solutions or providing examples."

I believe that this sentence should be followed by something like (authors should modify as they wish): "These differences may also be influenced by differences in geographical reviewer characteristics, quality of submissions, and the attitude of reviewers towards the journals". 

Otherwise the conclusion implies, at least to a degree, that higher impact may lead to more thorough reviews.

---

## [Editor Report · Decision Letter 3]

21 Jun 2023

Dear Dr Egger,

Thank you for your patience while we considered your revised manuscript "Relationship between Journal Impact Factor and the Thoroughness and Helpfulness of Peer Reviews" for publication as a Meta-Research Article at PLOS Biology. This revised version of your manuscript has been evaluated by the PLOS Biology editors and the Academic Editor.

Based on our Academic Editor's assessment of your revision, we are likely to accept this manuscript for publication, provided you satisfactorily address the following data and other policy-related requests.

IMPORTANT - please attend to the following:

a) Please ensure that you comply with our Data Policy; specifically, we need you to supply the numerical values underlying Figs 1, 2, 3, S1.1, S2.1, S2.2, S2.3, S2.4, and the 4 Figs in “S3 File”, either as a supplementary data file or as a permanent DOI’d deposition. We note that you mention a Zenodo URL (https://doi.org/10.5281/zenodo.8006829); however, this is not accessible, and I will need to see it before accepting the paper for publication.

b) Please cite the location of the data clearly in all relevant main and supplementary Figure legends, e.g. “The data underlying this Figure can be found in https://doi.org/10.5281/zenodo.8006829”

We expect to receive your revised manuscript within two weeks. 

*Published Peer Review History*

*Press*

Sincerely,

Roli Roberts

Roland Roberts, PhD

Senior Editor,

rroberts@plos.org,

PLOS Biology

DATA POLICY:

Regardless of the method selected, please ensure that you provide the individual numerical values that underlie the summary data displayed in the following figure panels as they are essential for readers to assess your analysis and to reproduce it: Figs 1, 2, 3, S1.1, S2.1, S2.2, S2.3, S2.4, and the 4 Figs in “S3 File.” NOTE: the numerical data provided should include all replicates AND the way in which the plotted mean and errors were derived (it should not present only the mean/average values).

DATA NOT SHOWN?

---

## [Editor Report · Decision Letter 4]

6 Jul 2023

Dear Dr Egger,

Thank you for the submission of your revised Meta-Research Article "Relationship between Journal Impact Factor and the Thoroughness and Helpfulness of Peer Reviews" for publication in PLOS Biology. On behalf of my colleagues and the Academic Editor, Ulrich Dirnagl, I'm pleased to say that we can in principle accept your manuscript for publication, provided you address any remaining formatting and reporting issues. These will be detailed in an email you should receive within 2-3 business days from our colleagues in the journal operations team; no action is required from you until then. Please note that we will not be able to formally accept your manuscript and schedule it for publication until you have completed any requested changes.

IMPORTANT: I note that you mention the reviewers ("Ludo Waltman, Bernd Pulverer and an anonymous reviewer") in the Acknowledgements. While we appreciate the sentiment, this is against PLOS policy, so please could you remove this? I will ask my colleagues to include this request in their list of issues to attend to.

Sincerely, 

Roli Roberts

Senior Editor

PLOS Biology

rroberts@plos.org